# Kiwifruit Resistance to *Sclerotinia sclerotiorum* and *Pseudomonas syringae* pv. *actinidiae* and Defence Induction by Acibenzolar-S-methyl and Methyl Jasmonate Are Cultivar Dependent

**DOI:** 10.3390/ijms242115952

**Published:** 2023-11-03

**Authors:** Tony Reglinski, Kirstin V. Wurms, Joel L. Vanneste, Annette Ah Chee, Magan Schipper, Deirdre Cornish, Janet Yu, Jordan McAlinden, Duncan Hedderley

**Affiliations:** 1Ruakura Research Centre, The New Zealand Institute for Plant and Food Research Limited, Hamilton 3214, New Zealand; kirstin.wurms@plantandfood.co.nz (K.V.W.); joel.vanneste@plantandfood.co.nz (J.L.V.); annette.ahchee@plantandfood.co.nz (A.A.C.); magan.schipper@plantandfood.co.nz (M.S.); deirdre.cornish@plantandfood.co.nz (D.C.); janet.yu@plantandfood.co.nz (J.Y.); jordan.mcalinden@outlook.com (J.M.); 2Palmerston North Research Centre, The New Zealand Institute for Plant and Food Research Limited, Palmerston North 4410, New Zealand; duncan.hedderley@plantandfood.co.nz

**Keywords:** *Actinidia*, MAMP, *Pseudomonas syringae* pathovar *actinidiae* (Psa), *Sclerotinia sclerotiorum*

## Abstract

Pathogen susceptibility and defence gene inducibility were compared between the *Actinidia arguta* cultivar ‘Hortgem Tahi’ and the two cultivars of *A. chinensis* ‘Hayward’ and ‘Zesy002′. Plants were treated with acibenzolar-s-methyl (ASM) or methyl jasmonate (MeJA) one week before inoculation with *Pseudomonas syringae* pv. *actinidiae* (Psa biovar3) or *Sclerotinia sclerotiorum*, or secondary induction with chitosan+glucan (Ch-Glu) as a potential pathogen proxy. Defence expression was evaluated by measuring the expression of 18 putative defence genes. ‘Hortgem Tahi’ was highly susceptible to sclerotinia and very resistant to Psa, whereas ‘Zesy002′ was highly resistant to both, and ‘Hayward’ was moderately susceptible to both. Gene expression in ‘Hayward’ and ‘Zesy002′ was alike but differed significantly from ‘Hortgem Tahi’ which had higher basal levels of *PR1-i*, *PR5-i*, *JIH1*, *NPR3* and *WRKY70* but lower expression of *RD22* and *PR2-i*. Treatment with ASM caused upregulation of *NIMIN2*, *PR1-i*, *WRKY70*, *DMR6* and *PR5-i* in all cultivars and induced resistance to Psa in ‘Zesy002′ and ‘Hayward’ but decreased resistance to sclerotinia in ‘Zesy002′. MeJA application caused upregulation of LOX2 and downregulation of *NIMIN2*, *DMR6* and *PR2-i* but did not affect disease susceptibility. The Ch-Glu inducer induced PR-gene families in each cultivar, highlighting its possible effectiveness as an alternative to actual pathogen inoculation. The significance of variations in fundamental and inducible gene expression among the cultivars is explored.

## 1. Introduction

Plants rely upon a combination of pre-formed and inducible defences to resist pathogen attack. Inducible defences are activated upon perception of attempted infection and play a critical role in plant pest and disease resistance [1]. All plants have inducible defences, but disease occurs when the defence response is too slow and/or weak to repel infection. Numerous studies have reported the use of different biological and chemical ‘inducers’ to activate inducible defences and enhance disease resistance in plants [2]. Inducers function by accelerating and amplifying the plant’s response to subsequent attack and by doing so can enhance the resistance capacity of individual cultivars without genetic modification or the need for time-consuming conventional breeding methods [2]. However, disease control resulting from induced resistance is often incomplete and inducer efficacy can be highly variable [3]. This variability is likely a function of several interconnected factors, including the speed of the initial recognition event, the complexity of crosstalk between antagonistic phytohormonal signalling networks, and the spectrum of the deployed defences [3,4,5]. A better understanding of these factors can help with optimization of inducer use and so reduce the variability of induced resistance for disease control. Furthermore, new knowledge of inducible defence may also provide an opportunity to harness the potential of induced resistance in breeding strategies and to develop cultivars with enhanced ‘inducer’ responsiveness.

The nature of host–pathogen interaction is crucial to the outcome of induced resistance for several reasons. Plants tend to activate different biochemical defence pathways according to the type of pathogen encountered [1]. In general terms, necrotrophic pathogens trigger the jasmonic acid (JA) pathway while the salicylic acid (SA)-dependent response is activated by biotrophs [6]. Moreover, disease resistance relies on a host response and enormous genomic variations within plant hosts, even down to the cultivar level in many horticultural crops, for example barley, apple, and tomato [7,8,9], to determine the specificity of the deployed defences ]. 

In this study we investigate defence gene expression in different kiwifruit cultivars following inducer application and subsequent pathogen inoculation. This crop was chosen as an exemplar because the commercial kiwifruit industry was initially dependent on just one cultivar but in the last 20 years production has been based around a relatively small number of genotypes [10], which have a high degree of genetic diversity [11,12]. Three commercially grown cultivars—the hexaploid, green-fleshed kiwifruit cultivar *Actinidia chinensis* var. *deliciosa* (A. Chev.) ‘Hayward’, the diploid, yellow-fleshed cultivar *A. chinensis* Planch. Var. *chinensis* ‘Zesy002′, also known as Gold3, and *Actinidia arguta* (Sieb. Et Zucc.) Planch. Ex Miq. ‘Hortgem Tahi’ (commonly referred to as kiwiberry)—were selected because of their different levels of basal resistance to a key necrotrophic and biotrophic pathogen [13,14,15,16,17]. 

The fungal necrotroph *Sclerotinia sclerotiorum* which causes sclerotinia disease is responsible for significant crop losses due to flower blight and fruit rot/scarring [14], and ‘Hortgem Tahi’ is highly susceptible to sclerotinia, ‘Hayward’ is moderately susceptible and ‘Zesy002′ is moderately resistant [13,14,15]. The bacterial biotroph *Pseudomonas syringae* pv. a*ctinidiae* (Psa) is considered today the most important global pathogen of kiwifruit [18]. Although *P. syringae* pv. *actinidiae* strains can be grouped into five different biovars, it is only strains of biovar 3 (Psa 3) which are responsible for the global outbreak of bacterial canker of kiwifruit [16]. Strains of Psa 3 affected cultivars of yellow-fleshed kiwifruit such as *A. chinensis* var. *chinensis* ‘Hort16A’ more strongly than the green-fleshed kiwifruit ‘Hayward’ [16]. However, the yellow-fleshed cultivar ‘Zesy002′, which was commercialized after the beginning of the global Psa outbreak, is less susceptible to Psa 3 than ‘Hayward’ and allowed the New Zealand kiwifruit industry to recover [16]. Strains of Psa 3 have been isolated from commercial orchards of ‘Hortgem Tahi’, but the economic impact of those outbreaks was negligible [17]. Unless they have a deletion in their exchangeable effector locus, strains of Psa 3 induce a hypersensitive response in *A. arguta* [19]. 

Acibenzolar-S-methyl (ASM) and methyl jasmonate (MeJA) were used as plant defence inducers in this study to activate the SA and JA defence pathways, respectively. ASM, an SA-mimic, has been shown to protect plants against a range of pathogens [20,21]. It is the active ingredient in Actigard^®^ which is used commercially in kiwifruit orchards to control Psa 3 because resistance to this pathogen is mediated by the SA pathway [22,23]. MeJA is a well-known inducer [24] and has been shown to upregulate JA gene expression in kiwifruit [25] and to induce resistance against a fungal necrotrophic pathogen of kiwifruit [26]. 

We also investigate the utility of a mixture comprising chitosan and β-glucan (Ch-Glu) as a potential proxy for pathogen-induced defence induction. The Ch-Glu treatment was considered to be a pathogen proxy because chitosan and β-glucan are conserved molecular features in the microbial cell walls, often referred to as microbe-associated molecular patterns (MAMPs) [27,28]. The recognition of MAMPs by surface receptors alerts the host plant to a potential threat and triggers a signalling cascade that promotes the activation of host defences [29]. Different receptors have been shown to recognise chitosan oligosaccharides [30,31] and β-glucans [32], and so combining the two to broadens its potential activity as a MAMP-based inducer of host defence and its capacity to function as a proxy for pathogen attack. The use of a proxy may be advantageous compared with pathogen challenge for time-course studies because there is a more defined time when the plant is exposed to the ‘defence trigger’. This is because pathogen recognition and infection following inoculation will be protracted, thus negating accurate determination of a specific time zero. Moreover, a proxy also avoids the need to culture the pathogen and so overcome challenges associated with quarantine pathogens that cannot be applied in the general environment. 

Marker genes selected for this study were defence genes shown to be involved in interactions of kiwifruit with biotrophic and necrotrophic pathogens, and/or key markers of phytohormone pathways in kiwifruit involved in defence/stress responses, especially the SA, JA and abscisic acid pathways [22,23,26,33,34,35].

This study was undertaken to better understand the relationship between inducer responsiveness and genotype. 

## 2. Results

### 2.1. Disease Susceptibility Is Cultivar Dependent

‘Hortgem Tahi’, ’Zesy002′ and ‘Hayward’ differ in susceptibility to sclerotinia and Psa 3, (disease symptoms shown in Appendix A). ‘Hortgem Tahi’ was the most susceptible cultivar to sclerotinia and developed larger leaf lesions (mean = 22 mm) in untreated controls than ‘Hayward’ (mean = 7 mm) or ‘Zesy002′ (mean = 1 mm) (Figure 1). In contrast, ‘Hortgem Tahi’ was the least susceptible to Psa 3 and expressed no measurable leaf necrosis whilst ‘Hayward’ was the most susceptible of the tested cultivars with approximately 14% leaf necrosis in untreated controls compared with less than 2% in ‘Zesy002′ (Figure 1). 

Foliar spray with ASM, one week before inoculation, did not affect sclerotinia infection in ‘Hortgem Tahi’ and ‘Hayward’ plants but increased susceptibility to sclerotinia in ‘Zesy002′ compared with untreated controls. Conversely, treatment with ASM induced a significant reduction in susceptibility to Psa 3 in ‘Zesy002′ and ‘Hayward’, compared with untreated controls. Treatment with MeJA did not significantly affect pathogen susceptibility (Figure 1). 

### 2.2. Gene Expression Is Specific to Cultivar Type and Inducer Compound 

To investigate the differences among cultivars and the effects of the inducer alone on basal and inducible gene expression, a pooled analysis of the pre-inoculation data from four experiments was performed. There was a significant cultivar effect for all genes except *AP2_ERF2* and *PR5-ii*, a significant inducer effect for 11 out of 18 genes, and a significant cultivar x inducer interaction only for *PR1-i* and *PR2-i* (Table 1).

Patterns of basal expression of defence genes for the two *A. chinensis* cultivars ‘Hayward’ and ‘Zesy002′ were very similar to each other, but different from that for *A. arguta* ‘Hortgem Tahi’ (Figure 2 and Appendix A). ‘Hayward’ and ‘Zesy002′ had extremely high levels of expression of *RD22* (approximately 170x higher than that in ‘Hortgem Tahi’), and moderately high levels of *RBOHF* (3× that in ‘Hortgem Tahi’) (Figure 2). In contrast, non-induced ‘Hortgem Tahi’ had significantly higher expression levels of a number defence genes—*PR1-i* (35× that of ‘Zesy002′ and 14× that in ‘Hayward’), *PR5-i* (11× that of ‘Zesy002′ and 4× that of ‘Hayward’), *JIH1* (6× that of ‘Hayward’/‘Zesy002′) *NPR3* (3× that of ‘Hayward’/’Zesy002′), and *WRKY70* (2× that of ‘Zesy002′ and 3x that in ‘Hayward’) (Figure 2). The elevated expression of several SA-related defence genes in non-induced ‘Hortgem Tahi’ relative to the *A. chinensis* cultivars corresponded with the greater resistance ‘Hortgem Tahi’ to Psa 3, in the absence of elicitation (Figure 1 and Figure 2). Basal expression of *PR2-i* was highest in ‘Zesy002′ (26× that of ‘Hortgem Tahi’ and 3× that of ‘Hayward’) and *BAD* was highest in ‘Hayward’ (20× that of ‘Hortgem Tahi’ and 3× that of ‘Zesy002′) (Figure 3). 

Averaged over cultivar, ASM significantly upregulated the same defence genes relative to the control treatment—*NIMIN2* (6×) *PR1-i* (5.9×), *WRKY70* and *DMR6* (3.8×), *PR5-i* (3.3×), *PR2-ii* (1.9×), *PR2-i* (1.8×), *PR1-ii* and *NPR3* (1.6×), and *PR3* (1.4×) (Table 1 and Figure 3). The only significant cultivar x inducer interactions were for *PR2-i*, and *PR1-i* to a lesser extent (Table 1). *PR2-i* was induced by ASM in ‘Hayward’ and ‘Zesy002′ but not in ‘Hortgem Tahi’, whereas induction of *PR1-i* occurred in all three cultivars but to a lesser extent in ‘Hortgem Tahi’, where the basal levels of expression were already high. This correlated with ASM-induced resistance to Psa 3 in ‘Hayward’ and ‘Zesy002′ (Figure 1).

Regardless of cultivar, MeJA significantly downregulated expression of several SA pathway-related genes—*DMR6* and *PR2-i* (0.5×), *NIMIN2* (0.7×) and *WRKY70* (0.8×) and upregulated expression of one JA pathway gene, *LOX2* (1.6×) (Figure 3) and had no statistically significant effect on resistance, although the use of MeJA did appear to increase Psa 3 infection in ‘Hayward’ (Figure 1).

### 2.3. Gene Expression Patterns Are Differentially Affected by Pathogen Inoculation and Oligosaccharide Challenge 

The largest fold-changes in response to inoculation with *S. sclerotiorum* (Figure 4) were: an upregulation of *AP2_ERF2* and *LOX2* in ‘Zesy002′; and upregulation of *PR5-ii* in ‘Hayward’. In addition, there were several genes upregulated in response to both mock and *S. sclerotiorum* inoculation—*AP2_ERF2*, *JIH1*, *PR1-ii*, *PR5-*i in ‘Zesy002′, *PR2-i* in ‘Hayward’ and ‘Zesy002′, and *LOX2* in all three cultivars. Overall ‘Zesy002′ had the strongest response to sclerotinia with five genes showing significant upregulation compared with the control. This corresponds with the greatest level of resistance to sclerotinia (Figure 1 and Figure 4). 

The largest fold-changes in response to Psa 3 inoculation (Figure 5) were: upregulation of *PR1-ii*, *PR2-i*, *PR2-ii*, *PR3*, *PR5-ii* and downregulation of *BAD* and *WRKY70* in ‘Hortgem Tahi’; upregulation of *AP2_ERF2*, *JIH1*, *PR2-*i and *WRKY70* in ‘Zesy002′; and downregulation of *NIMIN2,* and especially *PR5-ii* in ‘Hayward’. The upregulation of five genes in ‘Hortgem Tahi’ and four in ‘Zesy002′, along with significant downregulation of *NIMIN2* and *PR5-ii* in ‘Hayward’ is consistent with the Psa-resistance ranking of ‘Hortgem Tahi ‘> ‘Zesy002′ > ‘Hayward’ (Figure 1 and Figure 5). 

The application of an oligosaccharide mix (Ch-Glu) resulted in the upregulation of several genes in all cultivars (Figure 6). The most highly upregulated genes were *PR1-ii* and *PR2-ii* in ‘Hortgem Tahi’ and Zesy002′. Several genes were induced to a lesser extent, including *BAD*, *PR1-i*, *PR3* and *NIMIN2* in ‘Hortgem Tahi’; *BAD*, *PR2-i*, *PR5-ii* and *LOX2* in ‘Hayward’; and *PR3* in ‘Zesy002′.

### 2.4. Inducer Application Can Prime Host Gene Expression for Secondary Pathogen Challenge

Primary inducer application was a key determinant of gene expression patterns with ASM causing upregulation of most genes whilst, in contrast, MeJA mostly caused downregulation (Figure 7). However, there were cases where gene expression in both the mock and *S. sclerotiorum* inoculation treatments were greater than at 7d post ASM only. This was most evident in ‘Zesy002′ and suggests that ASM primed the plant to respond to the change in humidity that was imposed following inoculation to facilitate infection. In some cases, gene expression was greater following mock-inoculation than with *S. sclerotiorum* inoculation; examples include, *PR1-ii* and *PR2-i* in ‘Hortgen Tahi’, *PR5-ii* in ‘Hayward’, and most notably *AP2_ERF2*, *NIMIN2* and *PR1-i* in ‘Zesy002′, indicating a strong environmental response. For other genes, there was little difference between mock and *S. sclerotiorum* inoculation. There was less evidence of a priming effect to pathogen inoculation alone in ASM-treated plants. However, exceptions to this were an upregulation of *PR1-i*, *PR2-i* and *PR5-i* in ‘Hayward’ and of *DMR6* in ‘Zesy002′ following *S. sclerotiorum* challenge. There was also downregulation of *LOX2* after *S. sclerotiorum* inoculation in ASM-treated ‘Zesy002′. In MeJA-treated plants, the expression of *AP2_ERF2* and *LOX2* in ‘Hortgem Tahi’, and *PR1-i*, *PR1-ii*, *PR2-ii*, *PR3*, *PR5-i* and *RBOHF* in ‘Hayward’ were upregulated after *S. sclerotiorum* inoculation (Figure 7). The expression of *JIH1* decreased following *S. sclerotiorum* inoculation in ‘Hortgem Tahi’ in both ASM and MeJA pre-treated plants, as did *LOX2* in ASM-treated ‘Zesy002′. 

In the Psa inoculation study, treatment with ASM resulted in upregulation of most genes while the effect of MeJA was either neutral or a downregulation (Figure 8). The expression of *DMR6*, *NIMIN2* and *WRKY70* were induced by ASM in all cultivars; however, in general, gene upregulation by ASM was more evident in ‘Hayward’ and ‘Zesy002′ than in ‘Hortgem Tahi’ (Figure 8). Following inoculation with Psa 3, *NIMIN2*, *PR2-i*, *PR5-i*, *PR5-ii* and *RBOHF* were upregulated in ASM-treated ‘Hayward’. This corresponded with a significant induction of resistance against Psa 3 in this cultivar (Figure 1 and Figure 8). ‘Zesy002′ also expressed an ASM-induced resistance to Psa but only *PR5-i* was more strongly expressed following inoculation with Psa 3 than in mock-inoculated controls. (Figure 1 and Figure 8). In ASM-treated ‘Hortgem Tahi’ inoculation with Psa 3 resulted in downregulation of *AP2_ERF2*, *NIMIN2*, *JIH1*, *PR1-i* and *PR1-ii* compared with the mock-inoculated controls whilst *PR2-ii* and *PR3* were downregulated in both ASM and MeJA treated plants (Figure 8). However, changes in gene expression did not correspond with a significant measurable difference in resistance to Psa in this cultivar (Figure 1 and Figure 8). 

Gene expression patterns were measured following primary inducer application then secondary challenge with Ch-Glu, used as a proxy for pathogen inoculation (Figure 9). There was no pathogen inoculation and so plants were not placed in a humid environment to facilitate infection, hence no mock inoculation treatment. Gene expression was measured at 1 day (early response) and 7 days post-inducer (sampling time in pathogen studies) and at 1 day after Ch-Glu treatment (secondary challenge). Some genes were significantly upregulated at 1 d after primary induction but declined markedly by 7 days. Examples are *BAD* and *JIH1* in ASM-treated ‘Hayward’ and ‘Zesy002′, Most other upregulated genes in ASM-treated plants showed greater expression at 7 d than at 1 d post treatment. Early response genes in MeJA-treated plants included *BAD* and *PR5-i* in ‘Zesy002′, *PR1-ii*, *PR2-ii* and *PR3* in ‘Hortgem Tahi’ and *LOX2* in all three cultivars. 

Secondary challenge with Ch-Glu induced *PR1-i*, *PR2-i* and *PR5-i* in ASM-treated ‘Hayward’ to a similar or greater extent as observed following inoculation with sclerotinia (Figure 7) and Psa (Figure 8). Several other genes induced by Psa in ASM-treated ‘Hayward’, including *NIMIN2*, *PR2-ii*, *PR3*, and *WRKY70* were also induced by Ch-Glu. In ‘Hortgem Tahi’, the expression of *DMR6*, *WRKY70*, and *PR5-*I in ASM-treated plants were induced by both pathogens (Figure 7 and Figure 8) and by Ch-Gluc (Figure 9). Overall, gene expression levels were lower in MeJA-treated plants than in ASM-treated counterparts across all three cultivars. Nevertheless, secondary challenge of MeJA-treated plants with Ch-Glu induced an upregulation of *LOX2* in ‘Hortgem Tahi’ but caused downregulation in ‘Zesy002′. Similarly, the expression of *PR5-ii* was upregulated by Ch-Glu in MeJA-treated ‘Zesy002′ but downregulated in ‘Hortgem Tahi’ and ‘Hayward’. 

## 3. Discussion

Pathogen susceptibility and defence gene expression, before and after inducer application and challenge inoculation, were compared in the leaf tissue of the *A. arguta* cultivar ‘Hortgem Tahi’, the cultivar of *A. chinensis* var. *chinensis* ‘Zesy002′ and the cultivar of *A. chinensis* var. *deliciosa* ‘Hayward’. ‘Hortgem Tahi’ was highly susceptible to sclerotinia but was highly resistant to Psa whereas ‘Zesy002′ was moderately resistant to both, and ‘Hayward’ was susceptible to both. Commercially grown ‘Hortgem Tahi’ is highly resistant to infection by Psa biovar3 [17] and resistance may be associated with faster recognition of the pathogen [36] and activation of an effector-mediated hypersensitive resistance [19]. In this study, basal gene expression of SA–defence pathway genes was more highly expressed in ‘Hortgem Tahi’ than in ‘Zesy002′ and ‘Hayward’. Most notably, *PR1-i* was 35-fold greater and 14-fold greater, respectively, in ‘Hortgem Tahi’ than in ‘Zesy002′ and Hayward whilst *PR5-i* in ‘Hortgem Tahi’ was 11-fold greater than in ‘Zesy002′ and 4-fold greater than in Hayward. The SA–defence pathway plays a key role in kiwifruit resistance to Psa [22,23]. The elevated expression of SA-pathway genes in ‘Hortgem Tahi’ suggests that a strengthened innate immune response could precede the more pronounced effector-mediated response in this cultivar, as previously documented, compared to others, and fortify surveillance against biotrophic pathogens. 

Genes that were more highly expressed in ‘Zesy002′ and ‘Hayward’ than in ‘Hortgem Tahi’ included *RD22* (170×), *PR2-i* (>8×) and *RBOHF* (>3×). The expression of *RD22* is commonly associated with abiotic stresses such as drought [37] but its function in plant defence is poorly understood. The *PR2* gene family encodes hydrolytic enzymes (β-glucanases/β-glucosidases) that can degrade structural polysaccharides in fungal cell walls [38] and activate plant defence secondary metabolites through the removal of glycosyl conjugates [39]. The *RBOH* gene family encode for membrane proteins that function in plant development and stress responses including the production of reactive oxygen species associated with MAMP-recognition and stomatal closure [40,41,42]. Recent studies in rapeseed (*Brassica napus*) proposed a model whereby resistance to sclerotinia involves hydrolysis of the pathogen’s cell wall by host-derived glucanases and defence activation via RBOHF-oxidases [43]. The molecular basis of kiwifruit resistance to sclerotinia is poorly understood; however, it is possible that the stronger expression of *PR2* and *RBOHF* in ‘Zesy002′ and ‘Hayward’, compared with ‘Hortgem Tahi’, may partly explain their relative differences in resistance to foliar infection by *S. sclerotiorum.*

There is increasing interest in application of the plant defence inducer ASM for plant disease control but little investigation of potentially antagonistic effects on pathogens arising from defence pathway crosstalk. In the current study, the application of ASM, one week before pathogen inoculation, resulted in an induced resistance to Psa in ‘Zesy002′ and ‘Hayward’, but also a small but significant increase in susceptibility to sclerotinia in ‘Zesy002′. The upregulation of the SA––defence pathway genes observed in ASM-treated kiwifruit plants was consistent with previous studies [22,23,25] and may be associated with antagonism of JA-mediated defence pathways that play a role in resistance to necrotrophic pathogens like *S. sclerotiorum* [44]. Unfortunately, there were few markers of the JA–defence pathway in this study to determine whether downregulation of this pathway corresponded with sclerotinia disease, although one of the JA pathway markers genes used, LOX2, was downregulated in ASM-primed ‘Zesy002′ in response to *S. sclerotiorum* inoculation. Curiously, treatment with SA has previously been shown to induce foliar resistance to sclerotinia in kiwifruit [45], whilst in rapeseed both SA and MeJA were shown to induce resistance to sclerotinia [46]. Traditionally, *S. sclerotiorum* is viewed as a necrotroph, however, studies demonstrating sequential activation of SA and JA signalling pathways by this pathogen [46,47] suggest a more a complex lifestyle involving a two-phase infection process that switches from biotrophy to necrotrophy [44,48] and this was recently observed in kiwifruit [34]. In the current study, foliar application of MeJA did not affect sclerotinia susceptibility but did cause increased Psa leaf necrosis in ‘Hayward’, however, this was not statistically significant. Previous studies in ‘Hayward’ kiwifruit have shown that foliar application of MeJA increased leaf susceptibility to Psa leaf infection [49] and that increased susceptibility was concomitant with an increase in the endophytic Psa 3 population [33]. 

Gene expression analyses of inducer-treated plants revealed significant cultivar *x* inducer interactions for only *PR1-i* (*p* < 0.023) and *PR2-i* (*p* < 0.001). ASM induced the expression of *PR1-i* in all cultivars and was greatest in ‘Hortgem Tahi’ whereas *PR2-i* was significantly induced by ASM in ‘Hayward’ and ‘Zesy002′ but not in ‘Hortgem Tahi’. A pooled gene-expression analysis across all three cultivars showed that ASM induced significant increases in the expression of the SA––defence pathway genes *NIMIN2*, *WRKY70*, *DMR6*, *PR1-i*, and *PR5-i* in accord with its function as an SA-mimic. Interestingly, the expression of *PR1-i* and *PR5-i* in untreated ‘Hortgem Tahi’ was greater than in ASM-treated ‘Zesy002′ or ‘Hayward’, thus emphasising the cultivar difference. MeJA, as expected, induced significant upregulation of the JA-pathway marker *LOX2* and significant downregulation of SA-pathway markers *NIMIN2*, *DMR6*, and *PR2-i.* Generally, treatment with MeJA tended to cause downregulation in gene expression compared with controls but in most cases the reductions were not statistically significant. However, just as quantitative disease resistance relies on multiple minor additive gene effects [50,51] so cumulative effects of defence gene downregulation at a biological level should not be dismissed and, in this case, the potential effects on Psa leaf necrosis on MeJA-treated ‘Hayward’ leaves. Treatment of ‘Zesy002′ and ‘Hortgem Tahi’ with MeJA similarly caused an overall downregulation in gene expression but this was not associated with a significant effect on disease susceptibility.

The investigation of gene expression following inoculation was confounded because of similarities with responses to mock-inoculation in both the Psa and sclerotinia experiments. This may be indicative of a response to increased humidity (humidity tents), thus reflecting interplay between biotic and abiotic stress response-pathways. This could make sense from an evolutionary viewpoint whereby plants enter a state of readiness upon ‘recognition’ of conditions such as high humidity that favour pathogen infection [52]. In general, high humidity promotes the spread of bacterial and fungal growth; however, few studies have considered the effect of ambient humidity on the plant defence response [53]. In the current study, more SA–defence pathway genes (except *PR2-i*) were upregulated in response to both mock and Psa 3 inoculation in ‘Hortgem Tahi’ than in ‘Zesy002′ or ‘Hayward’, thus reflecting on possible diverse genetics associated with tolerance and resistance response to Psa in each of the polyploid cultivars [54,55]. Conversely, studies in *Arabidopsis* showed that the SA pathway (including *PR1*, *PR2* and *PR5*) was inhibited under high humidity and this was considered a significant determinant of enhanced susceptibility to Pst DC3000 [56].

In the sclerotinia trial, gene upregulation was greater in ‘Zesy002′ than in ‘Hayward’ and “Hortgem Tahi’, thus again reflecting relative levels of disease resistance. The response to inoculation in inducer-treated plants depended on the inducer with enhanced upregulation of SA-pathway genes in ASM-treated plants, compared with inoculated controls, and either a neutral response or downregulation of genes in MeJA-treated plants. The gene expression patterns tended to be similar for mock and pathogen inoculation within each experiment thus affirming the notion of response to high humidity and potential priming of this response by ASM. A more extensive time-course study is required to better differentiate between genes that respond to environmental change and those induced in response to pathogen infection. 

An oligosaccharide preparation, Ch-Glu, was evaluated in this study as a potential proxy for pathogen inoculation. It was considered that using a MAMP-type inducer avoids the need for specific environmental conditions and may mitigate some of spatiotemporal complications associated with pathogens, for example, the timing of infection relative to pathogen inoculation is neither predictable nor synchronised thus complicating interpretation of gene expression patterns. The application of Ch-Glu alone induced the expression of several genes including *PR1-ii* and *PR2-ii* genes in ‘Hortgem Tahi’ and ‘Zesy002′ and *PR2-i* and *PR5-ii* genes in ‘Hayward’. The JA-marker gene LOX2 was also induced in ‘Hayward’, thus indicating potential upregulation of both SA- and JA-mediated pathways and confirming the lack of specificity of the Ch-Glu inducer. Chitosan-oligosaccharides (COS) have proven effective as MAMPs in other studies and, in *Arabidopsis*, COS-induced resistance to *Pseudomonas syringae* pv. *tomato* DC3000 involved activation of both SA- and JA-mediated pathways [57]. The effect of secondary challenge with Ch-Glu on plants already treated with ASM or MeJA was difficult to determine because of strong primary inducer effects on gene expression. Interestingly, there was evidence of a priming response by ASM with further enhancement of *NIMIN2*, *PR2-i*, *PR2-ii*, *PR5-i* and *WRKY70* in ‘Hayward’ but not in the other cultivars. Curiously, the expression of several of these genes in ASM-treated ‘Zesy002′, including *PR1-i*, *PR2-i* and *PR5-i*, declined following Ch-Glu application but remained upregulated compared with the control. The differential cultivar response to Ch-Glu when used alone or when applied as a secondary challenge merits further investigation. 

In conclusion, this study found that basal gene expression patterns differed markedly between the kiwiberry ‘Hortgem Tahi’ and the two *A. chinensis* cultivars ‘Hayward’ and ‘Zesy002′. Pathogen resistance was also cultivar-dependent with ‘Hortgem Tahi’ demonstrating high susceptibility to sclerotinia but almost complete resistance to Psa, whereas ‘Zesy002′ exhibited moderate resistance to both, and ‘Hayward’ was moderately susceptible to both. Treatment with ASM induced resistance to Psa in ‘Zesy002′ and ‘Hayward’ but decreased resistance to sclerotinia in ‘Zesy002′. ASM induced a strong upregulation of SA-pathway genes in all three cultivars whilst MeJA either had no effect or caused gene downregulation. The high-humidity conditions imposed to favour infection also affected gene expression, thus confounding the identification of pathogen-specific responses. A Ch-Glu inducer activated genes in common with both Psa3 and *S. sclerotiorum*, thus indicating potential utility of this MAMP-type inducer as a proxy for pathogen challenge. However, these results in the current study are based on a limited number of genes and on a single tissue sample taken 1 d after pathogen/MAMP challenge and more extensive analysis is necessary. A broader set of SA and JA pathway genes and a more extensive time course would enable greater scrutiny of spatiotemporal aspects of the defence response. This would include more tissue sampling strategies to compare localized versus systemic responses [23], as well as the effects of tissue maturity and type. 

This study demonstrated the importance of comparing disease resistance and inducer-responsiveness at a cultivar level. Moreover, results showed potential consequences of phytohormonal crosstalk where upregulation of SA–defence pathways induced resistance to Psa but increased susceptibility to sclerotinia in ‘Zesy002′. This information is relevant to the kiwifruit industry because the number of commercially available cultivars is relatively small, and because ASM is widely used for control of Psa. Better understanding of inducer-responsiveness could facilitate the development of cultivar-specific disease control programs. Ch-Glu was identified as a potentially useful proxy for pathogen inoculation based on similarities in gene response patterns between Ch-Glu-treated plants and plants inoculated with S *sclerotiorum* and *Psa*. More extensive time-course studies across multiple genotypes and with a broader set of genes are warranted to better understand its potential utility. The availability of a pathogen proxy would facilitate selection of more ‘responsive’, and potentially more resistant, genotypes without the need for specific environments to encourage infection and/or containment facilities when handling quarantine microorganisms. 

## 4. Materials and Methods

### 4.1. Plant Material

Clonal tissue-cultured kiwifruit plants of *Actinidia arguta* ‘Hortgem Tahi’, *Actinidia chinensis* var. *deliciosa* ‘Hayward’ and *Actinidia chinensis* var. *chinensis* ‘Zesy002′ were obtained from Multiflora, Auckland, New Zealand. Individual plants were transplanted from the agar growth medium to 0.5 L planter bags filled to 2/3 with Daltons™ GB mix (Daltons, Matamata, New Zealand) and topped up with a 50:50 ratio mix of potting mix and perlite. The plants were placed in a glasshouse in high-humidity tents, with supplementary heating for the first 2 weeks of growth. Plants were further grown in the glasshouse (15–24 °C, 14 h day length) to about 30 cm tall, with at least 3–4 fully expanded leaves when used for experiments. A flood and drain system was used once daily to water the plants with a hydroponic nutrient solution, pH 6.2, containing calcium, iron, nitrates, sulphates, phosphates and trace elements (PGO Horticulture Limited, Tirau, New Zealand).

### 4.2. Inducer experiments

The SA pathway inducer Actigard^®^ (ASM, containing the active ingredient acibenzolar-S-methyl, Syngenta, Auckland, New Zealand) was prepared at a concentration of 0.2 g/L in Grade 3 Reverse Osmosis water containing a non-ionic organosilicone surfactant (0.05% *v/v* Du-Wett^®^, UPL New Zealand Ltd., Auckland, New Zealand) [DW water]. 

Methyl jasmonate (MeJA, Sigma-Aldrich, Inc. Auckland, New Zealand) was used to elicit the JA pathway and was prepared at a concentration of 2.25 mmol/L in DW water. Plants that were treated with MeJA were held in a well-ventilated area and spatially separated from the other treatments for 24 h to avoid cross contamination from volatile components. 

Control plants were sprayed with DW water. All treatments were applied using a handheld mist sprayer until just before leaf run-off and kept in the glasshouse for 7 days until pathogen inoculation (Section 4.3). A schematic outline of the experimental protocol in shown in Figure 10. 

There were five experiments (Exps) as outlined in Table 2. Gene expression by NanoString for Exps 1 and 2 were carried out simultaneously, and Exps 4 and 5 were analysed together on a separate occasion. 

### 4.3. Pathogen Preparation, Inoculation, and Disease Assessment

A mycelial plug from a pure culture of *S. sclerotiorum* (isolate Sc10-46), first isolated in 2010 from ‘Hayward’ floral tissue from an orchard in the Bay of Plenty, New Zealand, was sub-cultured onto potato dextrose agar (PDA) and incubated at 20–22 °C for 3 days. A mycelial plug from the growing edge of the colony was then sub-cultured onto PDA and grown for a further 3 days at 20–22 °C. Two fully expanded leaves per plant were each inoculated on their upper surface with two 5 mm agar plugs (mycelium side down) taken from the growing edge of the *S. sclerotiorum* culture. The plugs were placed mid-way between the midrib vein and the leaf edge. Plastic tents were placed over the plants to encourage infection. For mock inoculations, plants were inoculated with sterile agar plugs. 

*Pseudomonas syringae* pv. *actinidiae* biovar3 strain #10627 (Psa 3), first isolated in 2010 from *A. chinensis* var *chinensis* ‘Hort16A’ kiwifruit from Te Puke, New Zealand [23], was grown on King’s B medium [58] and incubated at 28 °C for at least 48 h. Bacterial colonies were re-suspended in sterile water and then the cell density was first estimated using a NovoSpec™ spectrophotometer (Amersham Biosciences, Auckland, New Zealand) (OD = 600 nm), then accurately determined by serial plate dilutions. The bacterial suspension was diluted with sterile RO water to give the final concentration of 2 × 10^7^ colony forming units (CFU)/mL and applied as an aerosol to the underside of kiwifruit leaves using a pressurized (40 psi) airbrush sprayer in a quarantine glasshouse. 

In Exps 1 and 2, the mock and pathogen inoculated plants were placed into a high-humidity tent (>90%) for 24 h after inoculation to facilitate infection.

For the disease assessments in Exp 3, the inoculated plants were placed into a tent for 48 h, after which the relative humidity was reduced to below 50% by opening two 20 × 20 cm flaps on top of the tent. The temperature inside the tent ranged from 18–32 °C. Sclerotinia disease severity was assessed 4 days after inoculation by measuring the leaf lesion diameters, based on the mean of four lesions per plant (2 per leaf × 2 leaves per plant) with 10 plants per treatment. For Psa, assessments of leaf necrosis (% leaf area) were made were made 14 days post-inoculation, based on the mean of three leaves per plant and 10 plants per treatment.

### 4.4. Oligosaccharide Preparation and Application

The oligosaccharide mixture (Ch-Gluc) was applied as an alternative to pathogen inoculation in Exps 4 and 5. This oligosaccharide inducer was a preparation of Armour-Zen^®^ (containing the active ingredient chitosan at 0.125 g/L) in combination with laminarin (at 0.125 g/L) in DW water. This was applied to the plants at 7 days after inducer treatment using a handheld mist sprayer until just before leaf run-off. The plants were not placed in high-humidity tents because there was no pathogen inoculation and therefore no need to facilitate infection. 

### 4.5. Leaf Sampling for Gene Expression Analysis

For Exps 1 and 2, leaves were sampled 7 days after inducer treatment (immediately before inoculation) and at 1 day after inoculation (both mock and pathogen inoculation)—i.e., 8 days after inducer treatment. Samples were collected from five plants per treatment by removing an interveinal section (3 cm × 1 cm) from two leaves per plant (~300 mg pooled sample). For leaves inoculated with *S. sclerotiorum*, the tissue sampled included the inoculation site. 

For Exps 4 and 5, leaves were sampled 1- and 7 days post inducer treatment and 1-day post Ch-Gluc application, i.e., 8 days post inducer treatment.

The leaf tissue was snap frozen in liquid nitrogen and then stored at −80 °C until processing. Only three of the five reps were processed for gene expression analysis. 

### 4.6. Gene Expression Analysis by PlexSet^®^ NanoString

#### 4.6.1. Selection of Genes 

PlexSet^®^ NanoString was used to provide direct counts of the number of genes of interest expressed within each sample and these counts were normalised against positive system controls and stably expressed reference genes (RG) [23,59]. The most stable RG and best differentially expressed genes of interest (GoI) from other studies looking at responsiveness of kiwifruit to inducer compounds [22,23,26,33,34,60] were used for Nanostring (Table 3). In some cases, more than one isoform was used from within a particular gene family because the different isoforms within each family have shown a differential response in various kiwifruit–microbe interactions [22]. Oligomer design for the RG and GoI probes was carried out by NanoString Technologies Inc. (Seattle, WA, USA), and were designed to be highly specific to the target sequence. The oligomer probes were synthesized by Integrated DNA Technologies Private Limited (IDT, Singapore) and comprise a reporter sequence, a capture sequence, plus probes A + B for 24 gene targets (including six reference genes) that hybridize to the target sequences of interest, forming a quadripartite complex.

#### 4.6.2. Sample Preparation

The total RNA from 100 mg samples of ground kiwifruit tissue was extracted using the Spectrum Plant Total RNA Kit (Sigma-Aldrich, Auckland, New Zealand) following the supplier’s recommendations. The RNA concentration and purity were determined by measurement of 1 µL of sample on a NanoDrop 200c spectrophotometer (Thermo Scientific, Waltham, MA, USA). The minimal acceptable concentration was 100 ng purified intact total RNA at ≥20 ng/µL. The RNA samples were sent on dry ice to Grafton Clinical Genomics (GCG, University of Auckland, Auckland, New Zealand), where RNA quality and concentrations were checked using a PipeJet^®^ Nanodispenser (BioFluidix GmbH, Freiburg, Germany) and then working aliquots of 50 ng/µL were prepared with nuclease-free water and stored at −80 °C.

#### 4.6.3. Titration Analysis to Determine the Optimal RNA Input (ng)

To determine the optimum RNA input for the PlexSet, a titration was run using three samples; two samples with expected high expression and one “Mix” sample (a pool of all samples to act as a calibrator for normalisation across different sets of barcodes). Each sample was run at different total RNA input amounts (50–350 ng). After an overnight hybridization of 19 h at 67 °C, total barcode counts were calculated and graphed on a scatter plot. The trend line equation was used to determine the optimal RNA input (ng) for 150,000 total counts, this being the optimum level as per Nanostring recommendations (MAN-10044-02, 2017). The resulting optimal RNA input of 150 ng was used in the first complete NanoString run for Exps 1 and 2, and 176 ng for the second Nanostring run for experiments 4 and 5.

#### 4.6.4. PlexSet Run 

For the PlexSet run, the 15 µL reaction volume in each well contained 5 µL hybridization buffer, 0.5 µL working probe A (0.6 nM each of Probe As), 0.5 µL working probe B (3 nM each of Probe Bs), 2 µL of the appropriate Plexset (A-H), 7 µL of sample RNA. The concentration of the RNA is calculated from the results of the initial titration run which determines the optimal total RNA input for independent assays. The plate was hybridized in the thermocycler at 67 °C for 19 h (GCGs standard hybridization times). Upon completion, the contents of the 96-well plate was pooled down into a 12-tube strip so that one tube contained all eight samples in the corresponding column and was then analysed by the Nanostring digital counter. 

#### 4.6.5. Gene Expression Data Analysis

Upon completion of the analysis, the results were uploaded into the nSolver^®^ software, version 4.0, provided by NanoString Technologies Inc. nSolver software was used to check: (1) counts associated with positive and negative controls (present in all tubes) to ensure that NanoString was amplifying correctly, without too much noise; (2) that the RGs were providing stable expression (checked by geNORM analysis), and; (3) that none of the data raised any Quality Control (QC) flags (no QC flags were present). Gene expression data were then normalised against the top three positive controls (highest counts); all the RGs (except for UBC and EF which were excluded from the normalisation as their coefficients of variation were high, indicating that they were not stably expressed); and the mixed sample, which was run across all eight PlexSets. Normalised data were then statistically analysed, using analysis of variance (ANOVA; Genstat 20th Edition). All expression data were log_2_-transformed to satisfy the assumptions of ANOVA (normal distribution and homogeneity of variance). Fisher’s least significant difference (LSD) at *p* ≤ 0.05 was used a means comparison test. 

### 4.7. Statistical Analysis of Disease Severity Data

Disease severity data were analysed as a randomised block design (RBD) by ANOVA. Data were log_2_-transformed to satisfy the assumptions of ANOVA and means separation by LSD (*p* ≤ 0.05), using Genstat 20th edition.

## 5. Conclusions

This study demonstrated the importance of comparing disease resistance and inducer-responsiveness at a cultivar level. ‘Hortgem Tahi’ was highly susceptible to sclerotinia and very resistant to Psa, whereas ‘Zesy002′ was highly resistant to both, and ‘Hayward’ was moderately susceptible to both. Gene expression in ‘Hayward’ and ‘Zesy002′ was alike but differed significantly with ‘Hortgem Tahi’ which had higher basal levels of SA–defence pathway genes, especially *PR1-i* and *PR5-i*, and the SA pathway is known to play a key role in kiwifruit resistance to Psa. Treatment with ASM (a SA mimic) caused upregulation of *NIMIN2*, *PR1-i*, *WRKY70*, *DMR6* and *PR5-i* in all cultivars and induced resistance to Psa in ‘Zesy002′ and ‘Hayward’ but decreased resistance to sclerotinia in ‘Zesy002′. This illustrates the potential consequences of antagonistic phytohormonal crosstalk between the SA– and JA–defence pathways, where upregulation of the SA-pathway induced resistance to Psa but also increased susceptibility to sclerotinia (which involves the JA–defence pathway) in ‘Zesy002′. MeJA application caused upregulation of LOX2 and downregulation of *NIMIN2*, *DMR6* and *PR2-i* but did not affect disease susceptibility. The Ch-Glu inducer induced PR-gene families in each cultivar, highlighting its possible effectiveness as an alternative to pathogen inoculation. This information is relevant to the kiwifruit industry because the number of commercially available cultivars is relatively small, and because ASM is widely used for control of Psa.

## Figures and Tables

**Figure 1 ijms-24-15952-f001:**
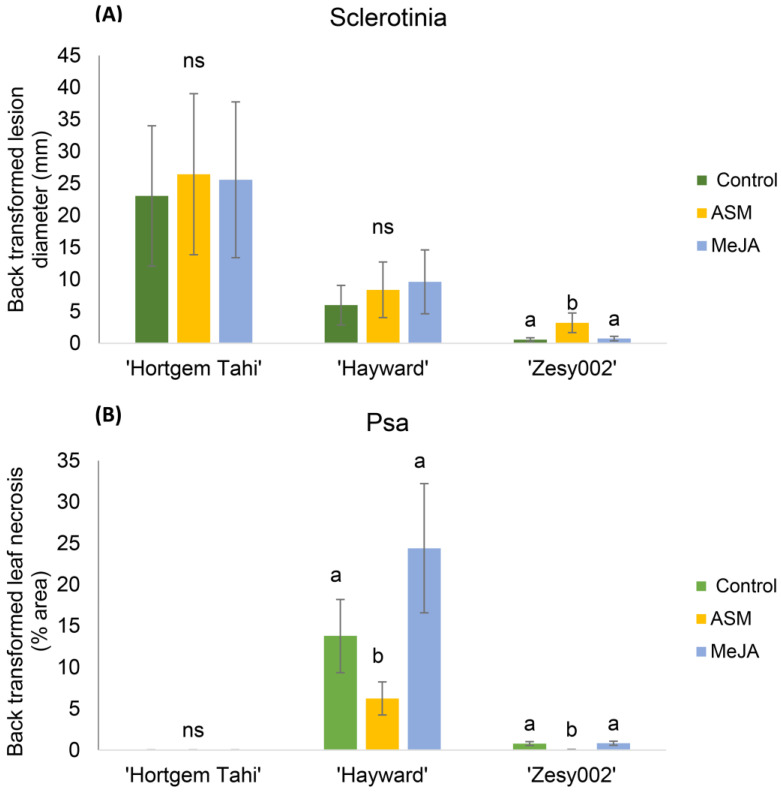
Relative susceptibilities of *Actinidia arguta* ‘Hortgem Tahi’, *A. chinensis* var. *chinensis* ‘Zesy002′ and *Actinidia chinensis* var. *deliciosa* ‘Hayward’ potted clonal kiwifruit plants to infection by (**A**) *Sclerotinia sclerotiorum* (sclerotinia), or (**B**) *Pseudomonas syringae* pv. *actinidiae* (Psa biovar3). Seven days before inoculation, plants were sprayed with either 0.2 g/L Actigard^®^ (ASM), or 2.25 mmol/L methyl jasmonate (MeJA) plus 0.05% *v/v* Du-Wett^®^ (DW) surfactant, or 0.05% *v/v* DW only (control). Disease assessments for Sclerotinia (lesion diameter) and Psa (% leaf area infected) were made 4 and 14 d after inoculation, respectively. Data were log_2_-transformed for analysis but are presented as back-transformed means. Error bars give the standard errors, and different lettering above the bars indicates significant treatment differences within cultivar comparisons, as shown by Fisher’s least significant difference (LSD), *p* ≤ 0.05. Non-significant differences are indicated by ‘ns’.

**Figure 2 ijms-24-15952-f002:**
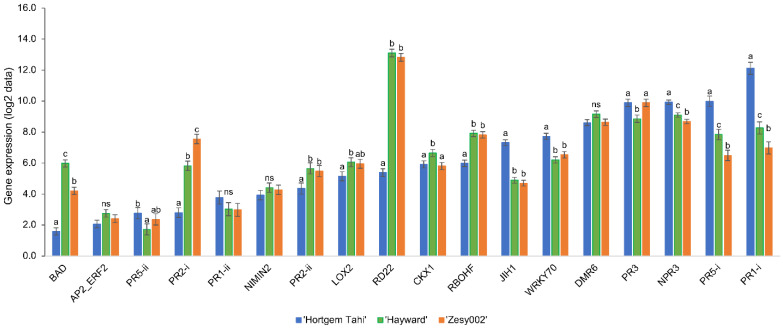
Cultivar effects on gene expression. Basal gene expression data (log2 gene counts) as measured by NanoString PlexSet^®^ in non-induced control plants of *Actinidia arguta* ‘Hortgem Tahi’, *Actinidia chinensis* var. *deliciosa* ‘Hayward’ and *A. chinensis* var. *chinensis* ‘Zesy002′ clonal plants that were sprayed with 0.05% *v/v* Du-Wett^®^ (DW) surfactant seven days before sampling. Data were pooled from all gene expression experiments and were log_2_-transformed for analysis after normalization against the reference genes 40s, Actin2, GAPDH and PP2A. Different lettering above bars indicates statistically significant differences as determined by LSD (*p* ≤ 0.05). NS = non-significant differences.

**Figure 3 ijms-24-15952-f003:**
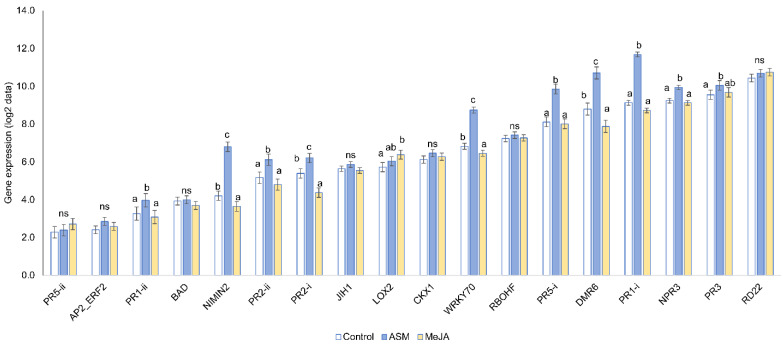
Inducer effects on gene expression. Gene expression data (log_2_ gene counts) as measured by NanoString PlexSet^®^ in kiwifruit plants that were sprayed with 0.2 g/L Actigard^®^ (ASM) or 2.25 mmol/L methyl jasmonate (MeJA) plus 0.05% *v/v* Du-Wett^®^ (DW) surfactant, or 0.05% *v/v* DW only (control) seven days before sampling. Data were pooled from all gene expression experiments, averaged across three cultivars and were log_2_-transformed for analysis after normalization against the reference genes 40s, Actin2, GAPDH and PP2A. Different lettering above the bars indicates statistically significant differences as determined by LSD (*p* ≤ 0.05). NS = non-significant differences.

**Figure 4 ijms-24-15952-f004:**
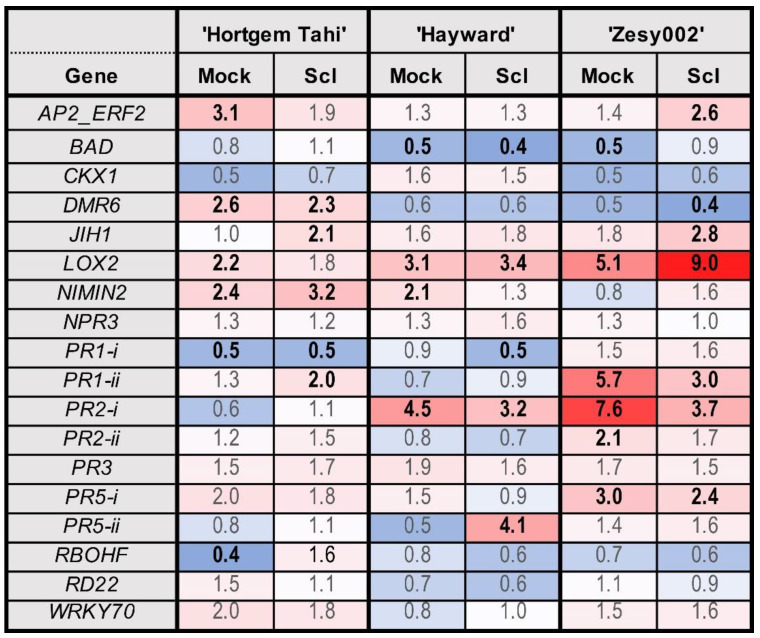
Heat map showing fold-change in gene expression in *Actinidia arguta* ‘Hortgem Tahi’, *Actinidia chinensis* var. *deliciosa* ‘Hayward’ and *A. chinensis* var. *chinensis* ‘Zesy002′ kiwifruit plants 1 day after inoculation with *Sclerotinia sclerotiorum* (Scl) or mock inoculation (with a sterile inoculum plug). Gene counts in leaf tissue were measured by NanoString PlexSet^®^ with normalization against the reference genes 40s, Actin2, GAPDH and PP2A. Fold-change data in red shades indicate an upregulation relative to the control, whilst blue shades represent downregulation relative to the control, with increasing colour intensity indicating larger fold-changes. Statistically significant differences from the control at individual time points, as determined by LSD (*p* ≤ 0.05), are identified in bold typeface.

**Figure 5 ijms-24-15952-f005:**
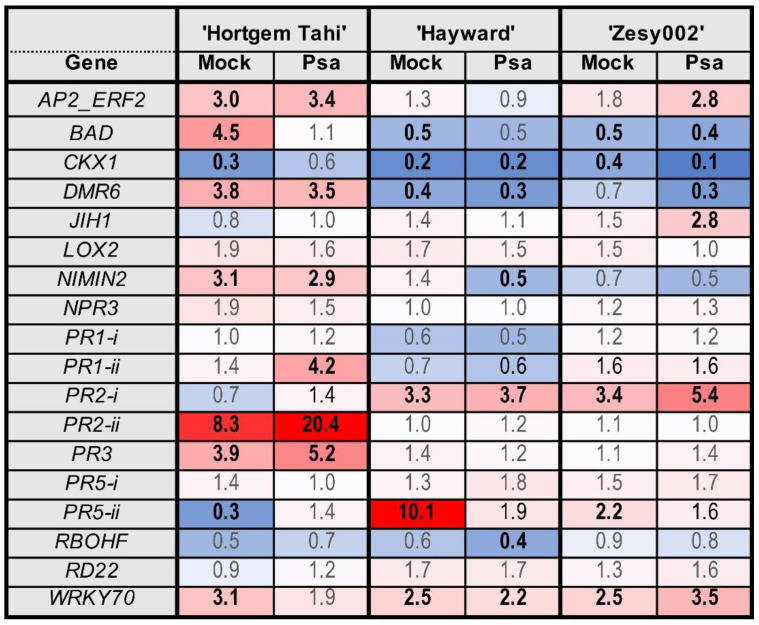
Heat map showing fold-change in gene expression in *Actinidia arguta* ‘Hortgem Tahi’, *Actinidia chinensis* var. *deliciosa* ‘Hayward’, and *A. chinensis* var. *chinensis* ‘Zesy002′ kiwifruit plants 1 day after inoculation with *Pseudomonas syringae* pv. *actinidiae* (Psa biovar 3) or mock inoculation. Gene counts in leaf tissue were measured by NanoString PlexSet^®^ with normalization against the reference genes 40s, Actin2, GAPDH and PP2A. Fold-change data in red shades indicate an upregulation relative to the control, whilst blue shades represent downregulation relative to the control, with increasing colour intensity indicating larger fold-changes. Statistically significant differences from the control at individual time points, as determined by LSD (*p* ≤ 0.05), are identified in bold typeface.

**Figure 6 ijms-24-15952-f006:**
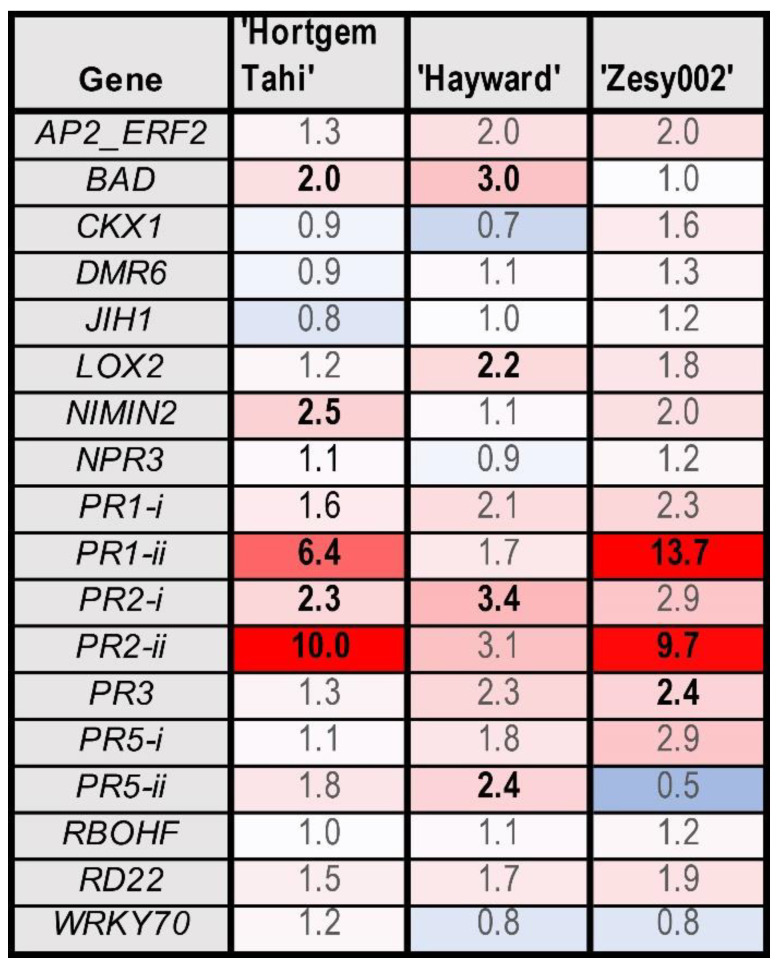
Heat map showing fold-change in gene expression in *Actinidia arguta* ‘Hortgem Tahi’, *Actinidia chinensis* var. *deliciosa* ‘Hayward’ and *A. chinensis* var. *chinensis* ‘Zesy002′ kiwifruit plants resulting from application of Ch-Glu (0.125 g/L each of chitosan and laminarin). Gene counts were measured by NanoString PlexSet^®^ with normalization against the reference genes 40s, Actin2, GAPDH and PP2A. Fold-change data in red shades indicate an upregulation relative to the control, whilst blue shades represent downregulation relative to the control, with increasing colour intensity indicating larger fold-changes. Statistically significant differences from the control at individual time points, as determined by LSD (*p* ≤ 0.05) are identified in bold typeface.

**Figure 7 ijms-24-15952-f007:**
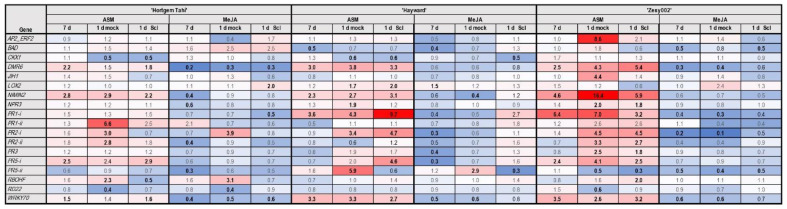
Heat map representing fold-change in gene expression in *Actinidia arguta* ‘Hortgem Tahi’, *A. chinensis* var. *chinensis* ‘Zesy002′ and *Actinidia chinensis* var. *deliciosa* ‘Hayward’ kiwifruit plants elicited by a foliar spray application of 0.2 g/L Actigard^®^ (ASM), or 2.25 mmol/L methyl jasmonate (MeJA) plus 0.05% *v/v* Du-Wett^®^ (DW) surfactant, or 0.05% *v/v* DW only (control), 7 days before inoculation with *Sclerotinia sclerotiorum* (Scl) or mock inoculation (with a sterile inoculum plug). Leaf samples were collected at 7 d post-inducer application, and 1 d post-challenge inoculation. Gene counts were measured by NanoString PlexSet^®^ with normalization against the reference genes 40s, Actin2, GAPDH and PP2A. Fold-change data in red shades indicate an upregulation relative to the control, whilst blue shades represent downregulation relative to the control, with increasing colour intensity indicating larger fold-changes. Statistically significant differences from the control at individual time points, as determined by LSD (*p* ≤ 0.05), are identified in bold typeface.

**Figure 8 ijms-24-15952-f008:**
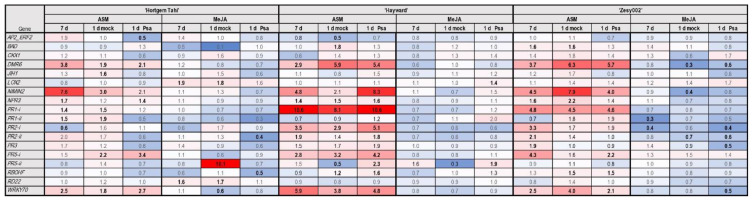
Heat map presenting fold-change in gene expression data in *Actinidia arguta* ‘Hortgem Tahi’, *A. chinensis* var. *chinensis* ‘Zesy002′ and *Actinidia chinensis* var. *deliciosa* ‘Hayward’ kiwifruit plants elicited by a foliar spray application of 0.2 g/L Actigard^®^ (ASM) or 2.25 mmol/L methyl jasmonate (MeJA) plus 0.05% *v/v* Du-Wett^®^ (DW) surfactant or 0.05% *v/v* DW only (control), followed by a challenge inoculation with *Pseudomonas syringae* pv. *actinidiae* (Psa biovar 3) or mock inoculation 7 days later. Leaf samples were collected at 7 d post-inducer application, and 1 d post-challenge inoculation. Gene counts were measured by NanoString PlexSet^®^ with normalization against the reference genes 40s, Actin2, GAPDH and PP2A. Fold-change data in red shades indicate an upregulation relative to the control, whilst blue shades represent downregulation relative to the control, with increasing colour intensity indicating larger fold-changes. Statistically significant differences from the control at individual time points, as determined by LSD (*p* ≤ 0.05), are identified in bold typeface.

**Figure 9 ijms-24-15952-f009:**
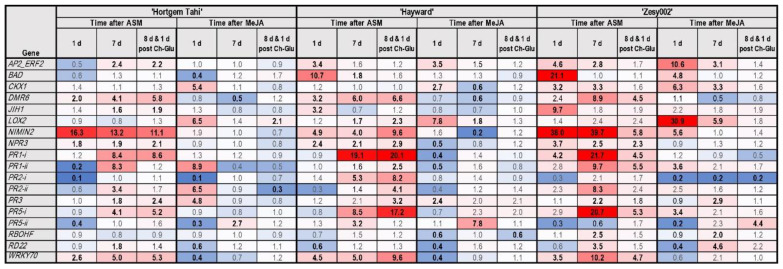
Heat map presenting fold-change in gene expression data in *Actinidia arguta* ‘Hortgem Tahi’, *A. chinensis* var. *chinensis* ‘Zesy002′ and *Actinidia chinensis* var. *deliciosa* ‘Hayward’ kiwifruit plants sprayed with 0.2 g/L Actigard^®^ (ASM), or 2.25 mmol/L methyl jasmonate (MeJA) plus 0.05% *v/v* Du-Wett^®^ (DW) surfactant or 0.05% *v/v* DW only (control), 7 days before treatment with Ch-Glu (0.125 g/L each of chitosan and laminarin). Gene expression was measured at 1 and 7 days post-inducer application, and 1 d post-Ch-Glu application (i.e., 8 days post-inducer application). Gene counts were measured by NanoString PlexSet^®^ with normalization against the reference genes 40s, Actin2, GAPDH and PP2A. Fold-change data in red shades indicate an upregulation relative to the control, whilst blue shades represent downregulation relative to the control, with increasing colour intensity indicating larger fold-changes. Statistically significant differences from the control at individual time points, as determined by LSD (*p* ≤ 0.05), are identified in bold typeface.

**Figure 10 ijms-24-15952-f010:**
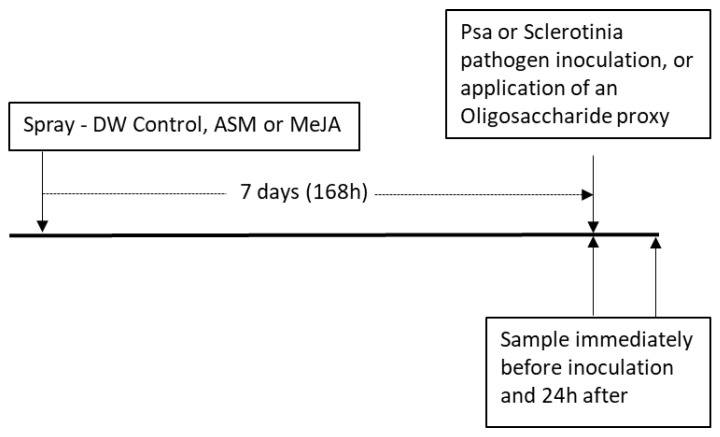
Schematic outline of the general experimental protocol and timeline. Original spray treatments consisted of spray application of distilled water (DW), Acibenzolar-S-methyl (ASM, 0.2 g/L) and methyl jasmonate (MeJA, 2.25 mmol/L). Pathogens used were *Sclerotinia sclerotiorum* (sclerotinia) and *Pseudomonas syringae* pv. *actinidiae* (Psa biovar 3).

**Table 1 ijms-24-15952-t001:** *p*-Values from the statistical analyses (ANOVA) comparing the effect of kiwifruit cultivar and inducer treatment on gene expression in *Actinidia arguta* ‘Hortgem Tahi’, *A. chinensis* var. *chinensis* ‘Zesy002′ and *Actinidia chinensis* var. *deliciosa* ‘Hayward’. Probability values ≤ 0.05 are indicated in bold.

Gene	Cultivar	Inducer	Cultivar × Inducer
*AP2_ERF2*	0.109	0.111	0.533
*BAD*	**<0.001**	0.326	0.884
*CKX1*	**0.043**	0.198	0.081
*DRM6*	**<0.001**	**<0.001**	0.687
*JIH1*	**<0.001**	0.095	0.167
*LOX2*	**<0.001**	**0.026**	0.712
*NIMIN2*	**0.041**	**<0.001**	0.274
*NPR3*	**<0.001**	**<0.001**	0.7
*PR1-i*	**<0.001**	**<0.001**	**0.023**
*PR1-ii*	**<0.001**	**0.028**	0.263
*PR2-i*	**<0.001**	**<0.001**	**<0.001**
*PR2- ii*	**<0.001**	**<0.001**	0.699
*PR3*	**<0.001**	**0.039**	0.392
*PR5-i*	**<0.001**	**<0.001**	0.314
*PR5-ii*	0.631	0.326	0.21
*RBOHF*	**<0.001**	0.545	0.692
*RD22*	**<0.001**	0.253	0.564
*WRKY70*	**<0.001**	**<0.001**	0.188

**Table 2 ijms-24-15952-t002:** Kiwifruit cultivar, inducer compounds and inoculation treatments in five experiments (Exp) on potted clonal plants of *Actinidia arguta* ‘Hortgem Tahi’, *Actinidia chinensis* var. *deliciosa* ‘Hayward’ and *Actinidia chinensis* var. *chinensis* ‘Zesy002′. Acibenzolar-S-methyl (ASM, 0.2 g/L) was used to elicit the salicylic acid pathway and methyl jasmonate (MeJA, 2.25 mmol/L) was the inducer for the JA pathway. Pathogens used were *Sclerotinia sclerotiorum* (Scl) and *Pseudomonas syringae* pv*. actinidiae* biovar *3* (Psa 3). The oligosaccharide mix, chitosan + laminarin (0.125 g/L of each) was used as an alternative to pathogen inoculation in Exps 4 and 5.

Exp #	Kiwifruit Cultivar	Inducer	Pathogen	Disease Assessment (n = 10 Plants)	Gene Expression (n = 5 Plants, with 2 Leaves/Plant)
1	‘Hayward’, ‘Zesy002′, ‘Hortgem Tahi’	ASM, MeJA, control	Scl	No	Yes
2	‘Hayward’, ‘Zesy002′, ‘Hortgem Tahi’	ASM, MeJA, control	Psa 3	No	Yes
3	‘Hayward’, ‘Zesy002′, ‘Hortgem Tahi’	ASM, MeJA, control	Scl, Psa 3	Yes	No
4	‘Hayward’, ‘Zesy002′	ASM, MeJA, control	Oligosaccharide proxy	No	Yes
5	‘Hortgem Tahi’ ^1^	ASM, MeJA, control	Oligosaccharide proxy	No	Yes

^1^‘Hortgem Tahi’ plants obtained at the same time as the ‘Hayward’ and ‘Zesy002′ plants in Exp 4, were not at the same stage of physiological development, so were assessed in a separate experiment (Exp #5).

**Table 3 ijms-24-15952-t003:** Identification of reference genes (RGs) and genes of interest (GoI) used for gene expression analysis by PlexSet^®^ Nanostring of kiwifruit defence responses in this study.

Identification (NCBI Entry/Acc#/Ach#)	Gene Name (Abbreviation)	Description/Function
FG526520/Acc31629.1 ^1^/Ach28g357801.1 ^2^	Elongation Factor (EF)	RG
FG498176/Acc23370.1 ^1^/Ach00g433851.2 ^2^	40s ribosomal protein (40s)	RG
FG478277/Acc01363.1 ^1^/Ach02g250301.2 ^2^	Ubiquitin-conjugating enzyme (UBC)	RG
FG499278/Acc18050.1 ^1^/Ach00g274261.2^2^	Glyceraldehyde 3-phosphate dehydrogenase (GAPDH)	RG
FG522516/Acc33246.1 ^1^/Ach29g419581.2 ^2^	Protein phosphatase 2A (PP2A)	RG
EF063572/Acc05529.1 ^1^/Ach05g107181.1 ^2^	Actin	RG
FG499230/Acc06864.1 ^1^/Ach06g146991.1 ^2^	Pathogenesis-related (PR) protein family 1, isoform i (PR1-i)	GoI—PR1 protein, a common salicylic acid (SA) pathway, induced by ASM and a kiwifruit-Psa resistance marker
Acc07445.1 ^1^/Ach07g240741.2 ^2^	PR1, isoform ii (PR1-ii)	GoI—PR1, common SA pathway marker, differential response to PR1-i
FG455092/Acc03929.1 ^1^/Ach03g327311.2 ^2^	PR2 glucan endo-1,3-β-glucosidase, isoform i (PR2-i)	GoI—PR2 protein, a common SA pathway, induced by ASM, and a kiwifruit-Psa resistance marker
FG480859/Acc23128.1 ^1^/Ach20g136371.2 ^2^	PR2 b-1,3-glucosidase, isoform ii (PR2-ii)	GoI—PR2 protein, a common SA pathway resistance marker, induced more strongly than isoform i by ASM in newly matured kiwifruit leaves and by Psa and by scale insects which induce the SA pathway
FG457667/Acc00338.1 ^1^/Ach01g209011.1 ^2^	PR3, class IV acidic chitinase (PR3)	GoI—PR3 protein, PR3 genes can be SA or JA pathway markers, but this gene is induced by ASM
AJ871175/Acc28854.1 ^1^/Ach25g132631.1 ^2^	PR5, thaumatin-like protein (TLP), isoform i (PR5-i)	GoI—PR5 protein, a common SA pathway, induced by ASM and kiwifruit-Psa resistance marker
FG417283/Acc28852.1 ^1^/Ach25g132651.2 ^2^	PR5, isoform ii (PR5-ii)	GoI—PR5 protein, a common SA pathway resistance marker, with differential expression vs. PR5-i (results in this paper)
Acc01173.1 ^1^/Ach00g239251.2 ^2^	Benzyl alcohol dehydrogenase (BAD)	GoI—a dehydrogenase with a preference for SA derivatives, induced by ASM in kiwifruit
Acc15406.1 ^1^/Ach00g326861.2 ^2^	Non-expressor of PR genes 3 (NPR3)	GoI—SA pathway marker that appears more relevant than commonly used NPR1 in kiwifruit, induced by ASM
Acc28514.1 ^1^/Ach25g374111.1 ^2^	NIM-interacting protein 2 (NIMIN2)	GoI—SA-induced NPR1-independent gene, possibly involved in regulation of PR1, induced by ASM in kiwifruit
Acc03241.1 ^1^/Ach03g045721.1 ^2^	Downy mildew resistance 6 (DMR6)	GoI—possible modulator of the SA response, induced strongly by ASM in kiwifruit
Acc03631.1 ^1^/Ach03g062591.1 ^2^	WRKY70	GoI—TF involved in SA pathway
Acc32487.1 ^1^/Ach28g123621.2 ^2^	Lipoxygenase (LOX2)	GoI—PR protein and a common jasmonic acid (JA) marker
Acc15135.1 ^1^/Ach13g018501.1 ^2^	Jasmonoyl-isoleucine-12-hydrolase (JIH1)	GoI—degrades the bioactive form of JA, increases in response to Psa inoculation
Acc20680.1 ^1^/Ach00g033321.1 ^2^	APETALA2 ethylene responsive factor 2 (AP2_ERF2)	GoI—transcription factor (TF) in the JA pathway
Acc29141.1 ^1^No Ach# match ^2^	Response to desiccation 22 (RD22)	GoI—marker of the abscisic acid (ABA) pathway
Acc09272.1 ^1^/Ach08g112681.2 ^2^	Cytokinin oxidase/dehydrogenase 1 (CKX1)	GoI—synthesis gene in the cytokinin pathway with sometimes decreases with excessive Actigard (ASM) use in kiwifruit (Wurms pers. comm.)—first published in this paper
Acc15636.1 ^1^/Ach14g188761.2 ^2^	Respiratory burst oxidase homolog gene F (RBOHF)	GoI—involved in the reactive oxygen species (ROS) burst response of kiwifruit to Psa

^1^ Acc. numbers come from the manual annotation of the Red5 kiwifruit whole Genome Shotgun project [61] that has been deposited at DDBJ/ENA/GenBank. ^2^ Ach. numbers replace the older Achn. numbering system and the sequences can be found at http://kir.atcgn.com, accessed on 20 September 2023.

## Data Availability

Data are contained within the article or the Appendix A.

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
