# Peer review of "Kiwifruit Resistance to Sclerotinia sclerotiorum and Pseudomonas syringae pv. actinidiae and Defence Induction by Acibenzolar-S-methyl and Methyl Jasmonate Are Cultivar Dependent"

_ijms, 2023, doi:10.3390/ijms242115952_

Round 1

Reviewer 1 Report

Comments and Suggestions for Authors

In this study, the authors focused on three different cultivars of kiwifruit, two pathogens with one of them is necrotrophic and the other is biotrophic, and two chemical regent acibenzolar-s-methyl (ASM) and methyl jasmonate (MeJA). the design is novel and the results are clear. My comments/suggestions are as follows:

1.      It would be better to show the disease symptoms together with the Figure 1.

2.      S. sclerotiorum can activate both SA and JA signaling, but the application of ASM significantly increases susceptibility to sclerotinia in ‘Zesy002’, is that because ASM application depresses the JA pathway?

3.      The authors tested many genes in the SA and JA pathway in each experiment, but why did not test the level of SA and JA? That will be more direct and convincing.

4.      How many replicates are there in the gene expression experiments? The author said five plants were used, but are these from one experiment? I mean no independent experiments were proceeded? As sometimes the gene level will be much too varied in the plant.

5.      In Figure 2 and Figure 3, error bars should be added.

Author Response

Thank very much for your comments - please see the attached file for our responses. 

Reviewer 2 Report

Comments and Suggestions for Authors

Kiwifruit resistance to Sclerotinia sclerotiorum and Pseudomonas syringae pv. actinidiae and defense induction by acibenzolar-S-methyl and methyl jasmonate are cultivar dependent

 The manuscript ijms 2664399 presents a research paper with relevant results established by properly executed tests. The analyses performed correspond to an exhaustive selection of factors to be taken into account for discussion.

 However, the following points should be provided or correctly explained:

 ·         Introduction

Line 35-Pg 59. It is necessary to include the scientific references of the data provided in both paragraphs, since they are the scientific background to justify the manuscript as well as the methodology carried out. The introduction lacks a correct justification of the work presented.

Line 60=line 87, it is the same sentence. The introduction must be correctly arranged

The results obtained should not be in the introduction

The justification for cultivar selection and its detailed description should be in the Materials and methods section.

·         Results

Figures in general: Avoid writing the methodologies carried out and already correctly described in materials and methods.

Fig. 1, 2 y 3: In the figures the results of significant differences should be presented in the same way. In the fig. 1, it is shown as asterisks and in Fig 2 and 3 as letters. Fig 2: No differences are shown as “aaa” and as “ns” in fig. 3.

Fig 1-A. “Sclerotinia”, write the acronym in Fig 1A as in the next figures (“Scl”) in order to facilitate the compression

·         Materials and methods

Line. 543, 552. A brief description of the origin and characterization of the Scl isolates and the Psa strain used in this work should be included.

A description of the methodology used to ensure that single-cell cultures have been used (if this has been done) and the techniques used to check their affiliation should be included.

It should be explained if, after inoculation of the plants, re-isolation or detection of the pathogens in the subsequently tested (infected) plants has been carried out.

Line 140-564. Discrepancies: “3 weeks after inoculation” or Fig. 1 “28 d post inoculation”

Line 504. It should be specified whether "Normal glasshouse conditions" corresponds to climate controlled conditions.

Line 555. The OD600nm measurement should be specified, which corresponds to the bacterial concentration 2x 107 CFU/mL.

·         Discussion

Line 491 and line 674. The results on which this conclusion is based should be better described in the ´results section´ and discussed.

·         References

Almost 40% of the references are from previous work by the authors and these are relevant to the varietal selection and the general issue of the manuscript, however, in the introduction, especially, references from other research teams, both in Kiwifruit and in other crops, should be included to justify the work and methodology carried out.

Author Response

Thank you for your review of our MS - please see the attached file for our responses.

Round 2

Reviewer 1 Report

Comments and Suggestions for Authors

I would like to thank the authors for addressing my questions.

The work presented in the current version is suitable to be published in IJMS and I have no more questions.